# Self-Supervised Adversarial Example Detection by Disentangled Representation

## Abstract

Deep learning models are known to be vulnerable to adversarial examples that are elaborately designed for malicious purposes and are imperceptible to the human perceptual system. Autoencoder, when trained solely over benign examples, has been widely used for (self-supervised) adversarial detection based on the assumption that adversarial examples yield larger reconstruction error. However, because lacking adversarial examples in its training and the too strong generalization ability of autoencoder, this assumption does not always hold true in practice. To alleviate this problem, we explore to detect adversarial examples by disentangled representations of images under the autoencoder structure. By disentangling input images as class features and semantic features, we train an autoencoder, assisted by a discriminator network, over both correctly paired class/semantic features and incorrectly paired class/semantic features to reconstruct benign and counterexamples. This mimics the behavior of adversarial examples and can reduce the unnecessary generalization ability of autoencoder. We compare our method with the state-of-the-art self-supervised detection methods under different adversarial attacks and different victim models (30 attack settings), and it exhibits better performance in various measurements (AUC, FPR, TPR) for most attack settings. Ideally, AUC is 1 and our method achieves $0.99+$ on CIFAR-10 for all attacks. Notably, different from other Autoencoder-based detectors, our method can provide resistance to the adaptive adversary.

## 1 Introduction

In 2013, the seminal work [1] reported that, during model test time, deep neural networks can be easily fooled by adversarial attacks that add tiny perturbation to inputs. Since then, adversarial attacks and defenses have drawn significant research attention [2–9]. On the one hand, attackers are persistently developing new strategies to construct adversarial examples; on the other hand, defenders are struggling to cope with all existing and forthcoming attacks [10].

Most of the existing defense methods [6–9, 11] are trained with supervision, and these methods work well when defending against adversarial attacks they were originally trained for. However, it is widely regarded that supervised methods cannot generalize well to adversarial examples from (existing) unseen attacks, let alone examples from new attacks.

Self-supervised based defense, in comparison with supervised defense, requires only benign examples for its training. As a typical example, the works in [12, 13] utilize the encoder of an autoencoder (AE) to draw the manifold of benign examples and then the decoder network for reconstruction, as shown in Fig. 1(a). Since the manifold is learnt from benign examples only and the AE is trained to minimize Reconstruction Errors (RE) for benign examples, thus the encoding of adversarial example will likely be out-of-distribution and the associated RE is larger.

Submitted to 35th Conference on Neural Information Processing Systems (NeurIPS 2021). Do not distribute.

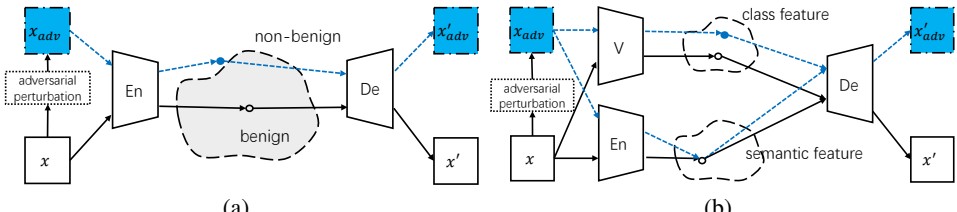

Figure 1: Different adversarial example detectors. Here, En stands for the encoder, De the decoder, V the victim model. (a) AE based method draws a manifold (the light gray area) of benign examples; (b) DRR disentangles images as two features: class features which represent images' labels (vulnerable to adversarial perturbation) and semantic features (robust to adversarial perturbation). When decoding class and semantic features, benign examples can be reconstructed faithfully but adversarial examples cannot.

It is soon realized that this is not always true because AE has very strong generalization ability [14]: examples with various kinds of small perturbations can be reconstructed with small RE. This is desirable if the perturbed examples are benign. However, adversarial examples are just specific perturbed versions of benign examples and the malicious perturbations can be also made very small in many attacks (i.e., now their encodings will reside in the light gray area of Fig. 1(a)). When this happens, all REs are mixed and it leads to high false negative or false positive rate (FNR/FPR) during detection. To refine the volume of the manifold drawn by AE and reduce its unnecessary generalization ability on adversarial examples, there exist a number of variants [15–18], as will be reviewed in detail in Sec. 2.2.

As a better solution to this problem, we propose a self-supervised disentangled representation based reconstruction (DRR) method to detect adversarial examples. DRR possesses the advantage of supervised defense, even though it does not have access to any adversarial examples in training. This is achieved through mimicking the behavior of adversarial examples by encoding and decoding a special class of examples (counterexamples in this work), which is the reconstruction of semantic feature from one example and class feature from an uncorrelated example. The rationale is based on the very fact of adversarial examples: **they cause misclassification without changing semantics** (contained in its benign counterparts).

With this fact, we disentangle, with the help of the victim model and an auxiliary encoder, the representation of images as two parts: class-dependent and class-independent (i.e., class feature and semantic feature). After disentangling, we then train a decoder to reconstruct benign examples by combining class/semantic features from the same benign image, as well as counterexamples by combining class/semantic features from different benign images (i.e., images with different labels). During detection, as shown in Fig. 1(b), DRR will faithfully disentangle any benign image to paired class/semantic features and its adversarial counterpart to unpaired class/semantic features, whose associated REs are significantly different as desired.

This paper makes the following contributions: 1) We use disentangled representation for adversarial detection, which makes it possible for the detector to mimic the behavior of adversarial examples in the self-supervised framework. 2) We design DRR via an AE structure, but it reduces the unnecessary generalization capability of AE on adversarial examples. 3) We achieve state-of-the-art adversarial detection performance on MNIST, Fashion-MNIST, and CIFAR-10 in most cases. Specifically, the Area Under Curve of Receiver Operation Characteristic (ideal AUC of ROC value is 1) is $0.99+$ on CIRFAR-10.

## 2 Background and Related Works

### 2.1 Constructing Adversarial Examples

The existence of adversarial examples in deep neural networks is first pointed out by [1], who find maliciously designed imperceptible perturbations can fool deep models to misclassify. Let $F(\cdot)$ be a general neural model and $C(\cdot)$ be the layers before $\mathrm{softmax}$ of the model, then evaluating the test example $x$ is simply a $\mathrm{softmax}$ classification over the logits $c = C(x)$, i.e., $y = F(x) = \mathrm{softmax}(c)$. The (untargeted) adversarial example $x_{\mathrm{adv}}$ derived from $x$ satisfies

$$x_{\mathrm{adv}} = x + \delta_{\mathrm{adv}}, F(x_{\mathrm{adv}}) \neq F(x), \text{ and } \|\delta_{\mathrm{adv}}\| < \epsilon, \tag{1}$$

where $\delta_{\mathrm{adv}}$ is the adversarial perturbation, $\epsilon$ is the maximum magnitude of $\delta_{\mathrm{adv}}$ under certain norm compliance. Commonly used norm could be $L_1$, $L_2$, and $L_{\mathrm{inf}}$, so we also focus on detecting adversarial examples bounded by these norms.

In the current literature, Fast Gradient Sign Method (FGSM) is the first widely used method in generating adversarial examples [2]. Basic Iterative Method (BIM) [19] and Projected Gradient Descent (PGD) [5] improve FGSM by iterating the building block of it according to different criteria to find the optimized perturbation. DeepFool [3] and CW [4] do not directly rely on gradient but they instead optimize (the norm of) $\delta_{\mathrm{adv}}$, which are usually more stealthy than FGSM and its variants.

Another line of research for constructing adversarial examples is called adaptive attacks [20]. Under an adaptive attack, the attacker also has white-box access to possible defense mechanisms, and his goal is to find an optimal $\delta_{\mathrm{adv}}$ that solves Eq. (1) and circumvents the defense mechanisms simultaneously. All the attacks above can have their adaptive versions by considering different defenses.

## 2.2 Detecting Adversarial Examples

Adversarial detection is an effective way to prevent adversarial examples, and it can be classified as supervised and unsupervised methods considering whether adversarial examples are needed for training. For the case of supervised detectors, their detecting capability depends on how to capture the differences between adversarial and benign examples. Techniques range from studying statistical properties [6–8], training traditional machine learning classifiers [21, 22] and deep classifiers [9, 11, 16, 23]. It is widely accepted that supervised methods cannot generalize well to adversarial examples produced by unseen attacks. For example, a supervised detector trained with PGD adversarial examples is likely to fail to detect examples produced by CW. However, it is also known that supervised detectors are robust to adaptive attacks, i.e., a detector trained with PGD will likely detect examples produced by the adaptive PGD attack [11, 23].

For the case of unsupervised detectors, their detecting capability depends on how to embed and represent benign examples to a different manifold (other than the natural spatiotemporal domain) thus adversarial perturbations will be magnified with embedding (without even seeing them at all). MagNet, proposed in [12], firstly uses an autoencoder to draw the manifold of benign examples, and it is regarded that the distance between the adversarial and benign examples are large via embedding and reconstruction.

However, the embedding manifold induced by autoencoder is not always desirable for detection, since autoencoder has too strong generalization capability [14]. Further to MagNet, the work [15] trains a variant of autoencoder by adding logits of the victim model into the loss function to refine the volume of the embedded manifold. The work [16] proposes to directly use parameters fixed victim model as the encoder, and the victim models' logits as high-level representations/embedding. Different from the works above which treat the manifold of all benign examples as a whole, [17, 18] propose a class-conditional model to embed and reconstruct benign examples for even better refining the (embedding) manifold. However, it is reported in [17] that this method still only reacts to relatively large (adversarial) perturbation over simple datasets. Compared with supervised detectors, self-supervised detectors may generalize better to unseen attacks but they are vulnerable to adaptive attacks [12, 14, 16]. For example, it is shown in [24] that the detector of [12] fails to resist adaptive attacks at all.

## 2.3 Disentangled Representation

Disentangled representation is firstly advocated by InfoGAN in [25], which encourages the learning of interpretable and meaningful representations of inputs for manipulating specific features. The work in [26] disentangles speaker-related representation to synthesis voices of different speakers. The work in [27] uses disentangled representation to address pose discrepancy problem among face images.

This paper uses disentangled representations for adversarial detection. It is pointed out [28] that there are many features in an image, and in adversarial attack, not all features are equally easy to be manipulated. Specifically, the class an image belongs to, which does not depend on all semantic features it has, is a concrete example. In this concern, we disentangle representations of an image

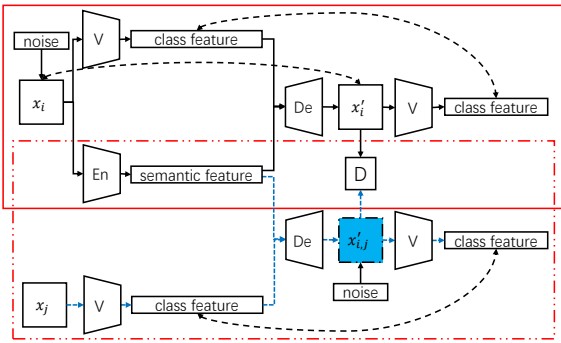

Figure 2: Overview of DRR.

into semantic feature and class feature. Semantic features can be easily manipulated by the attackers (through adversarial perturbation) but the perturbed features are similar to the original version (even after embedding if the perturbation is tiny enough). In contrast, class features cannot be directly manipulated (by attackers), but they are fragile to adversarial perturbation and robust to natural perturbation of the semantic features.

## 3 DRR for Adversarial Detection

This section presents the details of how to train DRR and use DRR to detect adversarial examples. For detection, as discussed earlier and shown in Fig. 1(b), an incoming test example $x$ will be encoded and decoded as $x'$, then the RE $\|x' - x\|_2$ is compared to a threshold value. If the RE is larger, then it is considered as adversarial; otherwise, it is not. Ideally, this threshold should be universal (on a given dataset) for all possible attacks (even for new attacks). However, for existing detectors, the threshold value is related to the used attack methods. This makes it hard to determine a universal threshold for existing detectors and to compare with them. For this reason, we use ROC curve of each attack strategy as the metric for evaluation in Sec. 4. But it is worth mentioning that it is easy to set a universal threshold for DRR for all the considered attacks.

We then move to how to train DRR. As depicted in Fig. 2, assisted by the parameter-fixed victim model $\mathsf{V}$, DRR first extracts class and semantic features from one benign example $x_i$ to train the encoder $\mathsf{En}$ and decoder $\mathsf{De}$. Then DRR takes the class feature from another benign example $x_j$ ($j \neq i$ and $x_j$'s label is different to $x_i$) and combines it with the semantic feature from $x_i$ to mimic the behavior of adversarial examples. The two encoding-decoding processes are further enhanced with the discriminator $\mathsf{D}$. The details of how to design the loss functions of the modular networks are presented in detail below.

First and foremost, we examine how to disentangle an input image $x$ to semantic feature $s$ and class feature $c$ for our detection purpose. For $s$, which represents the semantics of $x$, we use a general encoder network $\mathsf{En}$ to derive it, i.e., $s = \mathsf{En}(x)$. This is because it is widely accepted that high-dimensional input $x$ resides in a low-dimensional manifold, and it is a commonly used method in the area of representation learning. For $c$, which represents the class feature of $x$, we choose to use the logits from the victim model[1] $\mathsf{V}$, i.e., $c = \mathsf{C}(x)$. Aligned with the encoder-decoder structure and the discussion in Sec. 2.3, the rationale for this choice is two-sided: 1) Different from the one-hot encoding of the label of $x$, $\mathsf{C}(x)$ contains richer information for the subsequent decoding process; 2) Regardless of how the concrete adversarial example $x_{\mathrm{adv}}$ is constructed, $\mathsf{C}(x_{\mathrm{adv}})$ must be different enough from $\mathsf{C}(x)$ to induce the final erroneous classification.

With these features available, a decoder network $\mathsf{De}$ is then used to reconstruct $x'$ from $s$ and $c$ (of $x$), i.e., $x' = \mathsf{De}(s, c)$. The natural requirement for $\mathsf{En}$ and $\mathsf{De}$ is that, for a generic image $x_i$ from the benign dataset, the reconstructed version $x_i'$ should be similar to $x_i$. Moreover, we need to reserve the generalization capability over benign examples. For this purpose, an Gaussian noise vector $n_i$, each of the i.i.d. component follows $\mathcal{N}(0, \delta^2)$, is used for data augmentation, i.e., $\hat{x}_i = x_i + n_i$. Thus, the

---

[1]We hereinafter abuse the notation $\mathsf{V}$ in Fig. 2 to denote victim model without the final softmax layer as this will not cause ambiguity.

associated loss is

$$L_1 = \mathbb{E}_{x_i} \mathsf{MSE}(x_i, \hat{x}'_i), \tag{2}$$

where $\hat{x}'_i = \mathsf{De}\left(\mathsf{En}(\hat{x}_i), \mathsf{C}(\hat{x}_i)\right)$, MSE is the mean squared error and $\mathbb{E}$ is the expectation.

Different from the normal AE based detection and its variants, we also require the class features, after encoding and decoding of $x_i$ or its noisy version $\hat{x}_i$, are still aligned. So, another loss function associated with $x_i$ is

$$L_2 = \mathbb{E}_{x_i} \left[ \mathsf{MSE}(c_i, c'_i) + \mathsf{MSE}(\hat{c}_i, \hat{c}'_i) \right], \tag{3}$$

where $c_i = \mathsf{C}(x_i)$, $c'_i = \mathsf{C}(x'_i)$, $\hat{c}_i = \mathsf{C}(\hat{x}_i)$ and $\hat{c}'_i = \mathsf{C}(\hat{x}'_i)$. It is worth mentioning that we use the layer normalized version of $c_i/c'_i$ and $\hat{c}_i/\hat{c}'_i$ to calculate this cost in the experiment to avoid numeric instability and ease the task of hyperparameter tuning. This finishes our discussion of how to train En and De with paired semantic feature and class feature from benign examples (the red box with solid line in Fig. 2).

We then move to the training of DRR with unpaired semantic feature and class feature, as depicted by the red box with dashed line in Fig. 2. As emphasized in Sec. 2.2, the drawback of AE based detection is AE generalizes too well and the refinement of the manifold drawn by AE is not always effective, especially on attacks that directly optimize the norm of the adversarial perturbation. The solution is now straightforward since we can mimic the behavior of adversarial examples by constructing counterexamples from unpaired semantic feature and class feature to better refine the manifold drawn by En and De.

For another benign example $x_j$ with $j \neq i$, we extract its class feature via $c_j = \mathsf{C}(x_j)$. To obtain a counterexample, the semantic feature $s_i$ from $x_i$ and the class feature $c_j$ from $x_j$ are then passed to De to get $x'_{i,j} = \mathsf{De}\left(\mathsf{En}(x_i), \mathsf{C}(x_j)\right)$. Recall that our original purpose is to refine the manifold drawn by the encoder and decoder, so the loss function here should satisfy the following two requirements:

- The decoded $x'_{i,j}$ should not converge to $x'_i$ (and thus $x_i$) because convergence indicates that counterexamples and benign examples are mixed, which is contrary to our original detection purpose;
- The decoded $x'_{i,j}$ should not be far away from $x'_i$, since out-of-distribution examples do not really improve the detection capability of autoencoder.

For these reasons, we choose to use a soft hinge function as the loss, i.e.,

$$L_3 = \mathbb{E}_{x_i} \left[ \sum_{j, j \neq i} \max(0, d - \left\| x'_{i,j} - x'_i \right\|_2) \right], \tag{4}$$

where $d$ is a hyperparameter used to control the farthest allowable distance between counter and benign examples. This soft hinge function (passively) meets the second requirements list above: its gradient equals 0 when $\left\| x'_{i,j} - x'_i \right\|_2 > d$.

To further reduce the unnecessary generalization capability of the encoder-decoder network, we enforce class feature consistency over the counterexamples and their noisy versions. This is achieved by using the following loss function:

$$L_4 = \mathbb{E}_{x_i} \left[ \sum_{j, j \neq i} \mathsf{MSE}(c_j, \hat{c}'_j) \right], \tag{5}$$

where $\hat{c}'_j = \mathsf{De}(\hat{x}'_{i,j})$ with $\hat{x}'_{i,j} = x'_{i,j} + n_j$ (the component of $n_j$ also follows the Gaussian distribution $\mathcal{N}(0, \delta^2)$).

Inspired by [29], we treat the overall encoder-decoder (though assisted with the parameter-fixed victim model V) as a generator and then couple it with the last component of DRR, i.e., a discriminator network D. So (En+De) and D form the structure of generative adversarial networks (GAN), whose loss function is

$$
\begin{aligned}
L_{\mathsf{GAN}} = \; & \mathbb{E}_{x_i}[\log \mathsf{D}(x_i)] \\
& + \mathbb{E}_{x_i}[\log(1 - \mathsf{D}(\mathsf{De}(\mathsf{En}(x_i), \mathsf{V}(x_i))))] \\
& + \mathbb{E}_{x_i, x_j}[\log(1 - \mathsf{D}(\mathsf{De}(\mathsf{En}(x_i), \mathsf{V}(x_j)))))].
\end{aligned}
$$

The goal of this loss is to encourage the decoded examples, either benign or counterexamples, to be indistinguishable from the original input. It is worth mentioning that this GAN framework can be directly incorporated to all other encoder-decoder-like detectors [12, 13, 15–18], though they do not use a disentangled representation, to refine the manifold drawn by encoder-decoder networks.

In summary, the loss function to train the encoder En, the decoder De, and the discriminator D is

$$\text{Loss} = \sum_{i=1}^{4} \lambda_i \mathsf{L}_i + \lambda_{\text{GAN}} \mathsf{L}_{\text{GAN}}, \tag{6}$$

where $\lambda_i$ $(i = 1, \cdots, 4)$ and $\lambda_{\text{GAN}}$ control the relative importance of each loss function. En, De, and D are obtained by solving the minmax problem $\arg\min_{\text{En,De}} \max_{\text{D}}$.

The last observation we made is that even though the training of DRR mimics the behavior of adversarial examples, it does not really see any of them. And an empirical fact is that the victim model V is normally confident on benign examples. In contrast, it is not mandatory for V to be confident on adversarial examples (e.g., $\max(\text{softmax}(\mathsf{C}(x_{\text{adv}}))) = 0.2 << \max(\text{softmax}(\mathsf{C}(x))) = 0.9$). This leads to trivial failure of DRR as DRR relies on class feature $c = \mathsf{C}(x)$ for reconstruction. As a remedy of this problem, we use the class feature sharpening trick here, i.e., $c$ is updated as

$$c = \begin{cases} \alpha \cdot c - (\alpha - 1) \cdot \text{mean}(c), & \text{if } \text{std}(c) < \sigma; \\ c, & \text{others}, \end{cases} \tag{7}$$

where $\text{std}$ is the standard deviation function, $\sigma$ is the standard deviation of the training benign examples, and $\alpha$, an empirical sharpening constant, is set to 3.

Table 1: Detailed settings for the experiment.

| | MNIST Fashion-MNIST | CIFAR-10 |
|---|---|---|
| Optimization method | Adam | Adam |
| Learning rate | 0.0002 | 0.0002 |
| Training dataset size | 60K | 50K |
| Testing dataset size | 10K | 10K |
| Victim model | 8-layer CNN | VGG-16 |
| Encoder output size | 4 | 128 |
| $\lambda_1, \lambda_2, \lambda_3, \lambda_4, \lambda_{\text{GAN}}$ | 100, 1, 1, 3, 1 | 100, 1, 1, 3, 1 |
| $d, \delta$ | 0.5, 0.3 | 0.35, 0.1 |

## 4 Experimental Results

In this section, we assess the performance of DRR on three datasets, MNIST [30], Fashion-MNIST [31] and CIFAR-10 [32], by comparing it with other state-of-the-art detectors [12, 15, 16] against the adversarial attacks FGSM, BIM, PGD, DeepFool and CW (taken from Foolbox [33]) and the adaptive PGD under differnt norms ($L_1$, $L_2$ and $L_{\text{inf}}$). As a proof-of-concept, two representative networks, an 8-layer CNN and VGG-16 [34], are used as the victim models. These settings generate $(24 + 6)$ attacks and $(24 \times 5) + 6$ defenses, so only the representative results are reported here due to space limit and the complete results (including the source code) are provided in the supplementary file. We use a GeForce RTX 2080 to conduct the experiments.

### 4.1 Settings

**DRR**: DRR has three modules: encoder En, decoder De and discriminator D. It is suffice to say that the encoder En and the discriminator D have the same architecture as the victim model except the last fully connected layer: for CIFAR-10, there are 128 neurons in the last layer of En; for MNIST and Fashion-MNIST, only 4 neurons are used. The architecture of the decoder De is simply a mirror of En. Following the literature [11, 23], the hyperparameters $[\lambda_i]_{i=1}^{4}$ and $\lambda_{\text{GAN}}$ of the Loss are determined by a binary search in $[10^{-3}, 10^6]$, and $d$ and $\delta$ are determined empirically. The full details are listed in Table 1.

**Methods for comparison**: As mentioned above, three kinds of self-supervised detectors [12, 15, 16] are used for comparison. The structure of the encoder, the decoder for all these three methods are the same as of DRR. These methods can be classified as hidden vector based reconstruction (HVR) and high-level representation based reconstruction (HLR).

Table 2: Test accuracy of the victim models.

| Data | Attack | Accuracy |
|---|---|---|
| CIFAR10 | benign | 0.869 |
| | BIM $L_{\text{inf}}$ $\epsilon = 0.01$ | 0.195 |
| | BIM $L_1$ $\epsilon = 5$ | 0.588 |
| | PGD $L_{\text{inf}}$ $\epsilon = 0.01$ | 0.267 |
| | PGD $L_2$ $\epsilon = 0.1$ | 0.744 |
| | PGD $L_2$ $\epsilon = 0.3$ | 0.352 |
| | FGSM $L_{\text{inf}}$ $\epsilon = 0.05$ | 0.092 |
| | DeepFool $L_{\text{inf}}$ | 0.054 |
| | CW $L_2$ | 0.001 |
| MNIST | benign | 0.993 |
| | BIM $L_{\text{inf}}$ $\epsilon = 0.3$ | 0.001 |
| | BIM $L_1$ $\epsilon = 50$ | 0.012 |
| | PGD $L_{\text{inf}}$ $\epsilon = 0.3$ | 0.000 |
| | PGD $L_2$ $\epsilon = 1$ | 0.724 |
| | PGD $L_2$ $\epsilon = 2$ | 0.086 |
| | FGSM $L_{\text{inf}}$ $\epsilon = 0.1$ | 0.762 |
| | DeepFool $L_{\text{inf}}$ | 0.078 |
| | CW $L_2$ | 0.000 |
| Fashion | benign | 0.926 |
| | BIM $L_{\text{inf}}$ $\epsilon = 0.05$ | 0.021 |
| | BIM $L_1$ $\epsilon = 20$ | 0.300 |
| | PGD $L_{\text{inf}}$ $\epsilon = 0.05$ | 0.008 |
| | PGD $L_2$ $\epsilon = 1$ | 0.240 |
| | PGD $L_2$ $\epsilon = 2$ | 0.095 |
| | FGSM $L_{\text{inf}}$ $\epsilon = 0.05$ | 0.327 |
| | DeepFool $L_{\text{inf}}$ | 0.052 |
| | CW $L_2$ | 0.003 |

Table 3: AUC of ROC of different detectors over different datasets.

| Data | Norm | HVR-P | HLR-P | HVR-L | HLR-L | DRR (ours) |
|---|---|---|---|---|---|---|
| CIFAR10 | BIM $L_{\text{inf}}$ $\epsilon = 0.01$ | 0.4755 | 0.6340 | 0.4766 | 0.9039 | **0.9992** |
| | BIM $L_1$ $\epsilon = 5$ | 0.4380 | 0.5996 | 0.4364 | 0.8515 | **0.9975** |
| | PGD $L_{\text{inf}}$ $\epsilon = 0.01$ | 0.4704 | 0.6199 | 0.4723 | 0.8856 | **0.9990** |
| | PGD $L_2$ $\epsilon = 0.1$ | 0.4411 | 0.5992 | 0.4469 | 0.8568 | **0.9988** |
| | PGD $L_2$ $\epsilon = 0.3$ | 0.4594 | 0.6105 | 0.4613 | 0.8742 | **0.9983** |
| | FGSM $L_{\text{inf}}$ $\epsilon = 0.05$ | 0.6620 | 0.7000 | 0.6430 | 0.8207 | **0.9977** |
| | DeepFool $L_{\text{inf}}$ | 0.5051 | 0.6518 | 0.5038 | 0.8167 | **0.9954** |
| | CW $L_2$ | 0.5020 | 0.6562 | 0.5019 | 0.8298 | **0.9982** |
| MNIST | BIM $L_{\text{inf}}$ $\epsilon = 0.3$ | 0.4711 | 0.8509 | 0.3972 | 0.8672 | **0.9939** |
| | BIM $L_1$ $\epsilon = 50$ | 0.7572 | 0.8876 | 0.7391 | 0.8968 | **0.9714** |
| | PGD $L_{\text{inf}}$ $\epsilon = 0.3$ | 0.5226 | 0.8291 | 0.4350 | 0.8551 | **0.9970** |
| | PGD $L_2$ $\epsilon = 1$ | 0.2313 | **0.9897** | 0.2633 | 0.9886 | 0.9691 |
| | PGD $L_2$ $\epsilon = 2$ | 0.1645 | 0.9948 | 0.2149 | 0.9915 | **0.9977** |
| | FGSM $L_{\text{inf}}$ $\epsilon = 0.1$ | 0.0724 | 0.9352 | 0.0811 | **0.9569** | 0.9367 |
| | DeepFool $L_{\text{inf}}$ | 0.3824 | 0.9653 | 0.4301 | 0.9748 | **0.9815** |
| | CW $L_2$ | 0.7211 | **0.9871** | 0.6947 | 0.9828 | 0.9750 |
| Fashion | BIM $L_{\text{inf}}$ $\epsilon = 0.05$ | 0.5255 | 0.8295 | 0.5389 | 0.8419 | **0.8880** |
| | BIM $L_1$ $\epsilon = 20$ | 0.5420 | 0.8311 | 0.5481 | 0.8419 | **0.8944** |
| | PGD $L_{\text{inf}}$ $\epsilon = 0.05$ | 0.5267 | 0.8439 | 0.5415 | 0.8555 | **0.9000** |
| | PGD $L_2$ $\epsilon = 1$ | 0.5614 | 0.9610 | 0.5728 | 0.9662 | **0.9858** |
| | PGD $L_2$ $\epsilon = 2$ | 0.6972 | 0.9654 | 0.7259 | 0.9709 | **0.9914** |
| | FGSM $L_{\text{inf}}$ $\epsilon = 0.05$ | 0.6113 | **0.9201** | 0.6298 | 0.9193 | 0.9192 |
| | DeepFool $L_{\text{inf}}$ | 0.6137 | 0.9088 | 0.6270 | 0.9273 | **0.9378** |
| | CW $L_2$ | 0.5068 | 0.8981 | 0.5013 | 0.9058 | **0.9125** |

For [12], it is based on optimizing the pixel-level reconstruction error from the hidden vector outputted by the encoder. We term it as HVR-P. For [15], it optimizes both pixel and logits reconstruction error and we term it as HVR-L. For [16], it takes the logits from the parameter-fixed victim model as a high-level representation of examples for decoding, and it has two variants, HLR-P and HLR-L, depending on whether the loss function includes the logits reconstruction error or not. We also incorporate and optimize the GAN loss when implementing all these detectors: HVR-P, HVR-L, HLR-P, and HLR-L. Note that our implementations perform slightly better than the original methods in [12, 15, 16], but this provides a fair environment for evaluating the effectiveness of disentangled representation.

## 4.2 Results Analyses

With the settings above, we first train victim models on the 3 datasets, and then generate adversarial examples with the 5 attacks (FGSM, BIM, PGD, DeepFool, and CW) under different norm compliance ($\epsilon$ in Eq. (1)). All these test examples, either benign or adversarial, are passed to the detector for inspection.

**Preparing test examples**: The two victim models, VGG-16 and 8-layer CNN, are trained on CIFAR-10 and MNIST/Fashion MNIST, respectively. For each test example, we generate the corresponding adversarial versions with 5 attacks under different norm compliance, and the test accuracy is listed in Table 2. From this table, it is clear that the victim models achieve 0.869, 0.926, 0.993 classification accuracy over benign test examples. Also from this table, the attacks DeepFool and CW, which directly optimize $\epsilon$, are generally stronger than the other three. By increasing $\epsilon$, other attacks also obtains better attack result (i.e., worse accuracy), e.g., PGD with different $L_2$ constraints in Table 2. However, a larger adversarial perturbation will make the attack get detected easier.

**Visual inspection**: We visually compare the reconstructed results of the benign and adversarial examples for all the detectors, some examples are shown in Fig. 3. Inspecting columns 3 and 4 of Fig. 3(a) and (b), it is clear that DRR, HLR-P and HLR-L generally have bigger reconstruction errors over adversarial examples, which validates that logits (of the victim model) do help to reduce undesirable generalization capability of the detectors.

We further box-plot the reconstruction errors of all benign and adversarial examples from CIFAR-10, the result is shown in Fig. 4. It is clear from this figure, under all attacks of CIFAR-10, the reconstruction errors of DRR are clearly separable. The reconstruction errors of HLR-P and HLR-L are also generally separable, but it has a higher FNR or FPR than that of DRR (depending on the concrete threshold of RE). This validates that the disentangled semantic and class features can better refine the volume of the embedded manifold and prevent undesirable generalization. Moreover, as shown in this figure, it is easy to determine a universal threshold for DRR to detect all considered

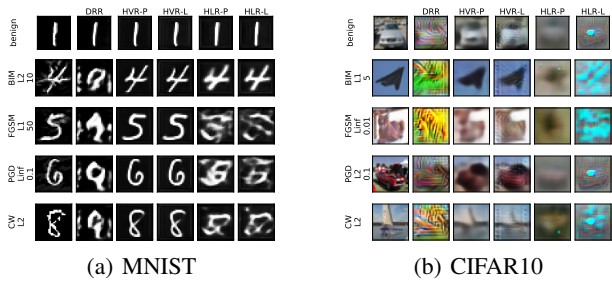

(a) MNIST            (b) CIFAR10

Figure 3: Reconstructed images of each detection methods.

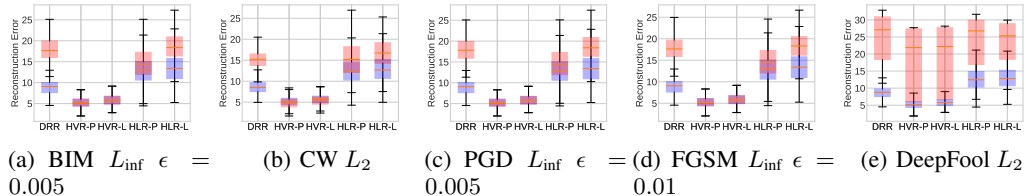

(a) BIM $L_{\inf}$ $\epsilon$ = 0.005    (b) CW $L_2$    (c) PGD $L_{\inf}$ $\epsilon$ = 0.005    (d) FGSM $L_{\inf}$ $\epsilon$ = 0.01    (e) DeepFool $L_2$

Figure 4: Box-plot of the reconstruction errors for CIFAR-10 (red box is for adversarial and blue box is for benign).

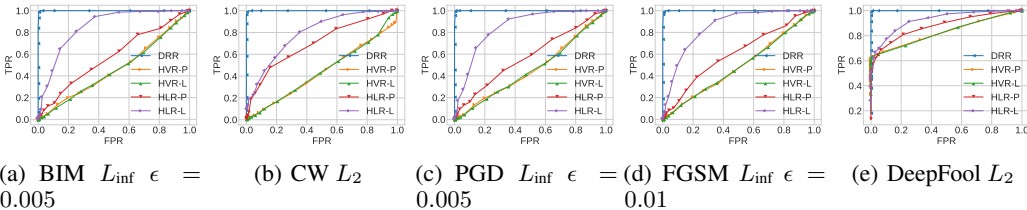

(a) BIM $L_{\inf}$ $\epsilon$ = 0.005    (b) CW $L_2$    (c) PGD $L_{\inf}$ $\epsilon$ = 0.005    (d) FGSM $L_{\inf}$ $\epsilon$ = 0.01    (e) DeepFool $L_2$

Figure 5: ROC curve of all detectors under various attacks on CIFAR-10.

attacks over CIFAR-10 (and it is similar for the other two datasets), but it is hard to do so for HVR-P, HVR-L, HLR-P, and HLR-L.

**ROC curve and AUC**: With benign samples and adversarial examples of CIFAR-10, the ROC curves of all the detectors are depicted in Fig. 5 for qualitative analysis. From Fig. 5, it is very clear that DRR can retain a high true positive detection rate (TPR) while keeping the FPR rate low. If the size of the AUC of ROC curve is larger, the detector is better and the ideal size can reach up to 1.

We use AUC of ROC to quantitatively study the effectiveness of all detectors over CIFAR-10, as well as extending this metric to MNIST and Fashion-MNIST. The results are tabulated in Table 3. It is clear that, for CIFAR-10, the AUC of DRR is the largest over all different attacks, and the value is very close to the ideal case 1. For MNIST and Fashion-MNIST, DRR outperforms other detectors in most cases. For the cases that DRR is inferior to HLR-P/HLR-L, the gap is tiny and we regard their performances are comparable to each other.

### 4.3 Defending Against Adaptive Adversarial Attack

To further evaluate the performance of DRR, we assume that the attacker can not only access the victim model but also knows all the details of the detector. Under this adaptive assumption, the attacker's goal is to fool both the victim model and the detector. Following the most-widely used adaptive attack strategy [8, 11, 20, 23, 24], the attack now aims to solve

$$\min_{x_{\mathrm{adv}}} \quad \alpha \mathrm{L_{RE}}(x_{\mathrm{adv}}) - \mathrm{L_{CE}}(x_{\mathrm{adv}}, y)$$
$$\text{s.t.} \quad \|x_{\mathrm{adv}} - x\|_p < \epsilon, \ \ p \in \{1, 2, \inf\} \quad (8)$$

where $\mathrm{L_{RE}}$ and $\mathrm{L_{CE}}$ are respectively the attack's loss function for the detector (i.e., reconstruction error) and for the victim model (i.e., cross entropy), and $y = \mathsf{F}(x)$ is the true label of $x$ and $\alpha$ is the

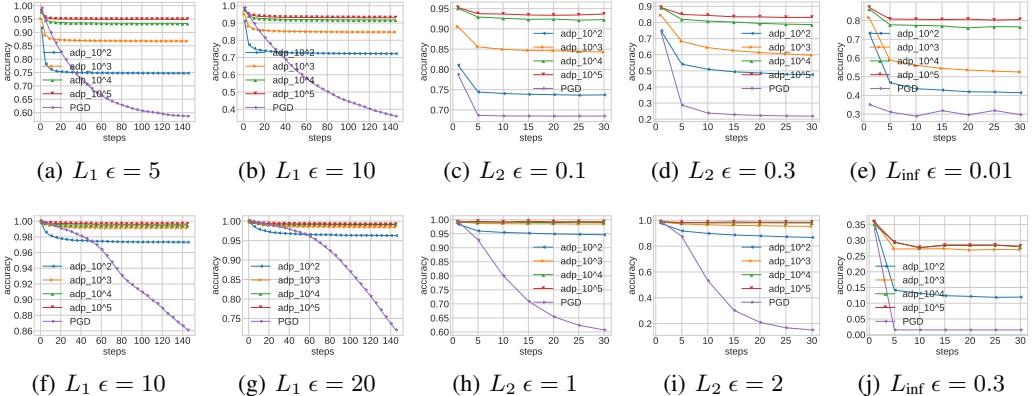

(a) $L_1\ \epsilon = 5$  (b) $L_1\ \epsilon = 10$  (c) $L_2\ \epsilon = 0.1$  (d) $L_2\ \epsilon = 0.3$  (e) $L_{\text{inf}}\ \epsilon = 0.01$

(f) $L_1\ \epsilon = 10$  (g) $L_1\ \epsilon = 20$  (h) $L_2\ \epsilon = 1$  (i) $L_2\ \epsilon = 2$  (j) $L_{\text{inf}}\ \epsilon = 0.3$

Figure 6: Accuracy of victim model under the original PGD attack and the adaptive attack with different $\alpha$. The first row is for CIFAR10, the second row is for MNIST.

parameter to control the relative importance of the two loss functions. It is worth mentioning that the detectors [12, 15, 16] in comparison fail to resist the adaptive attack defined above[2]. The reason is, for other AE-based detectors, no matter high-level representation (i.e., logits) is used or not, they fail to capture true features of adversarial examples by certain. In contrast, the AE in DRR mimics the behavior of adversarial examples via disentangled representation.

We evaluate this adaptive attack strategy under the same setting of Table 1, and the results for CIFAR-10 and MNIST are depicted in Fig. 6. Observing the first row of Fig. 6, it is clear the accuracy of victim model under adaptive attacks is higher than that without our defense, a.k.a., it is harder for the attacker to succeed when the detector DRR is deployed. Moreover, if the attacker focuses more on circumventing the detector by increasing $\alpha$ of Eq. (8) (from $10^2$ to $10^5$), it will be harder for him to succeed in constructing *real* adversarial examples. However, if the attacker puts less focus on bypassing the detector, further experimental results verify that it makes him easier to get detected.

A similar trend can be also observed when implementing the adaptive attack on MNIST (the second row of Fig. 6). But from this figure, it is clear that now it is harder for the attacker to succeed in attacking, for example, the attacker only has about $10\%$ attack success rate (under $L_1$ and $L_2$ norm) even with small $\alpha = 10^2$ (less attention on circumventing the detector). We regard this is because the semantic feature of MNIST is simple, and the network architecture used by DRR (Table 1) captures this feature very well and the defense is stronger (than that of CIFAR-10 which has more complex semantic features). In this concern, it is reasonable to speculate that DRR can be extended to other complex datasets with an appropriate choice of model architecture to capture the complex semantic features within the dataset, which we leave as future work.

## 5 Conclusion

In this study, we propose to make use of disentangled representation for self-supervised adversarial examples detection. The proposed method DRR is based on the very nature of adversarial examples: misleading classification results without changing images' semantics (a lot). With disentangled class and semantic features, this nature inspires us to construct counterexamples to better guide the training of DRR. Compared with previous self-supervised detectors, DRR generally performs better under various measurements over different datasets and different adversarial attack methods. Not surprisingly, compared with other AE-based adversarial detectors, DRR is also more robust to adaptive adversaries. This makes DRR a promising candidate, when combined with other proactive strategies, for the defense of adversarial attacks. Moreover, disentangled representation is a ubiquitous property for many other data formats, including natural language, voice, and video signals, so DRR may be extended easily to provide adversarial defense to other domains.

---

[2]We note that [15] considered a weaker adaptive attack strategy (maximize the cross entropy after AE reconstruction but not directly optimize reconstruction error) and HVR-L is robust to the weaker adaptive notion.

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
