# OpenReview forum: "Self-Supervised Adversarial Example Detection by Disentangled Representation"
_NeurIPS.cc/2021/Conference — NeurIPS 2021 Submitted_

### Official Review · Reviewer_aJ6q · 2021-06-27

**Rating:** 5
**Confidence:** 3

**Summary:**

This paper proposes to detect adversary using learned disentangled auto-encoder. The model can disentangle the class feature and the semantic feature. Class features are easily changed by adversary, while the semantic features cannot. Existing Auto-Encoder cannot disentangle those two types of features, and also has strong generalization ability---high performance on both the adversary and the benign examples, making it hard to detect. Using the proposed disentangled AE, benign example can be reconstructed faithfully while adversary examples cannot. Thanks to the tailored AE, the paper state-of-the-art adversarial detection performance on MNIST, Fashion-MNIST, and CIFAR-10 in most cases.

**Limitations And Societal Impact:**

 The authors adequately addressed the limitations and potential negative societal impact of their work

**Main Review:**

Pros:
1. The paper is well motivated by the fact that AE generalize well even under adversary attacks, and propose a novel design to the AE, which disentangles the class features and semantic features, to allow detection of adversary.

2. The paper evaluated on three datasets, MNIST, Fashion-MNIST, and CIAFR-10, over 5 different adversarial attacks. The gain is significant.

3. It is nice to see the method is evaluated under adaptive attacks. The adaptive attacker attacks the detection metric in an end-to-end manner, which is a standard adaptive attack. However, it may be weak due to attacking an auto-encoding task that is high dimension, making the attack less effective, because gradient from multitask (one task for each pixel reconstruction) may cancel out [1].

4. Overall, the paper is clear written.

Cons:
1. Adaptive attack maybe weak. As I mentioned before, auto-encoding task is not a task that is easy to attack due to its high output dimensionality. Thus the author should consider a stronger adaptive attack that attack one component in the model that is less high dimension. For example, instead of attack End2Eend, we can attack the individual component of the defense model, if we have that knowledge.  What if the adaptive attacker attacks(optimizes, corrects) the V model that encode the class feature so that it is the same as the natural label? In this way, the detector should not longer working because class feature is the same as the benign.

[1] Mao et al. Multitask Learning Strengthens Adversarial Robustness. ECCV 2020.



**Time Spent Reviewing:**

2

---

> ### Author Response · Authors · 2021-08-10
> **Response to reviewer aJ6q**
>
> ## Q1. Attack the component of the defender's model as a stronger attack.
>
> The reviewer is correct in noting that we are adopting the standard adaptive adversarial attack. We further remark this is the only known adaptive attack.
> In the adaptive attack, the assumption is that the defense network needs to be treated as an integrated algorithm/network model/box, and there are no adaptive attacks that focus on just one or few components within the box.
> This is analog to our real-life scenarios, an old-fashioned lock is a mechanical fastening device and the key has notches that correspond to the obstructions in the lock, we all know these facts but we don't study the security of a lock by breaking it into pieces.
> Moving to the stronger adaptive attack suggested by the reviewer, if the attacker can break the defense network into components and just work on individual component one-by-one, then it is easier for the attack to achieve his goal by directly breaking the victim model into components and freely manipulate the component that is in charge of final output.

---

### Official Review · Reviewer_kDjj · 2021-07-16

**Rating:** 6
**Confidence:** 3

**Summary:**

The paper proposes a novel adversarial example detection strategy based on disentangled latent representations. The main idea is that latents are separated into class features and semantic features. Detection then uses the fact that adversarial samples preserve semantics while at the same time changing class according to some victim model. Training is being performed through a sophisticated combination of losses accounting for begin examples and counterexamples. Experimental results on detecting various kinds of attacks on popular datasets show clear improvements over previous methods. The paper also presents an evaluation of an adaptive attack.

**Ethical Concerns:**

None.

**Limitations And Societal Impact:**

The paper includes an adaptive attack in which white-box access to the presented defense mechanism is granted. It would have been great if the paper would also discuss additional limitations of the proposed method (see questions above).

**Main Review:**

- Originality: The paper presents a novel combination of common ideas to learn disentangled representations for adversarial sample detection. While previous works in this area have investigated traditional (non-disentangling) autoencoders. Related work seems to be sufficiently citied and discussed.

- Quality: The submission appears technically sound and well organized. The method and individual losses appear well motivated and the experimental evaluation is solid.

- Clarity:
The paper is generally clearly written. The presence of a few missing connecting words and typos, these generally do not harm the reading flow. However, I found the methodological section a bit harder to follow at first. One suggestion might be to include an example similar to Figure 2 but using images from CIFAR-10 or MNIST instead of x_i or x_j. I also had the impression that the text in the adaptive evaluation was a hastily written and could need a bit of rework to increase clarity.

- Significance: The experimental setup is sound and the presented results are noticeably better compared to competing approaches. I especially found the AUROC curves impressive. The approach is also reasonably resistant to the adaptive attack presented in the paper.

- Questions:
1) I would be curious to get further insights into the disentanglement quality. Would it be possible to add further results to the paper that visualize disentanglement quality?
2) It is apparent from Figure 3 (b) that DRR does not reconstruct the benign sample well while 1-2 other methods do a reasonable job at this task. Granted that faithful reconstruction is not the main task, I wonder whether the authors could comment on this tradeoff between reconstruction quality of benign (!) samples and detectability of adversarial examples a bit more. At the same time, I understand that  the constructed loss is aimed at reducing the generalization capability of the autoencoder to more easily detect adversarial samples.

**Time Spent Reviewing:**

3

---

> ### Author Response · Authors · 2021-08-10
> **Response to reviewer kDjj**
>
> ## Q1. Further insights into the disentanglement quality.
> ## Q2. Clarity the trade-off between reconstruction quality of benign examples and detectability of adversarial examples.
>
> For Q1, as we replied to the reviewers above, our method cannot disentangle an image into disentangled features and this is not our goal. But instead, DRR can make good use of disentangled image features (the label information from the parameter-fixed victim model is called the class feature, and the representation from the encoder is called the semantic feature) for adversarial detection. The label and content of an image are already disentangled quite well: if everyone calls dogs cats, then dogs will be cats from now on though their appearances are not changed.
>
> For Q2, we find that increasing the quality of reconstruction for benign examples cannot improve detection. The detection ability of all autoencoder-based detectors is based on the gap in reconstruction errors between benign examples and adversarial examples. In our work, this gap is guaranteed by $d$ in the hinge loss $L_3$. If $d$ is too small, then the reconstruction errors of benign examples and adversarial examples are mixed and the detectability of adversarial examples becomes bad. If $d$ is large, the upper bound of the allowable loss (given by Eq. (6)) is increased even if all the other hyper-parameters remain the same, then the allowable reconstruction error for benign examples is also increased (indirectly), which explains that why DRR does not reconstruct benign example very well in Fig. 3(b).

---

> > ### Comment · Reviewer_kDjj · 2021-08-30
> > **Thanks for the clarifications**
> >
> > I thank the authors for the clarifications!
> >
> > Q1) I see your point. After reading through your discussion with reviewer 8oh2, I agree that this part of the paper should be re-written to emphasize the type of disentanglement in order to avoid ambiguity with feature entanglement.
> >
> > Q2) This makes sense! It would be great if this discussion could be added to the paper since this might appear to be an important tradeoff.
> >
> > I will keep my score unchanged.

---

### Official Review · Reviewer_xJBn · 2021-07-18

**Rating:** 6
**Confidence:** 4

**Summary:**

The paper propose a new defense mechanism against adversarial attacks based on an autoencoder architecture. The proposed architecture uses self-supervised learning to disentangle the semantic and class features, which enables the autoencoder to discriminate the clean and adversarial examples at test time. The approach was shown to outperforms other autoencoder approach in the experiments, when the attacker as access to only the victim model or both the victim model and autoencoder.

**Limitations And Societal Impact:**

Yes

**Main Review:**

**Originality**:
- The proposed approach builds on previous autoencoder approach to detect adversarial examples, however the author propose a new architecture and leverage self-supervised learning.
0The related work seems to be adequately cited and the contributions is put into the context of the previous autoencoder approach.

**Quality**:
- The experiments clearly show that the proposed approach outperforms the other autoencoder approach. In particular, table 3 clearly shows that DRR has a significantly higher AUC on 3 datasets (CIFAR, MNIST, Fashion) and against different attackers. However, the experiments don't report any confidence interval, it would be nice to have error bars for the experiments.
- The objective is also composed of many different losses and introduces many new hyperparameters, the role and importance of each losses is not clear, it would be nice to have some ablation studies that shows to which extent each loss terms and how the new hyperparameters affects the performance of the proposed solution. In particular, I don't understand the definition of $L_2$ the first term is a constant and $\hat x_i$ is not defined. In eq. 4 how important is the hinge loss and the value of $d$. I would like to see an ablation study with $\lambda_{\text{GAN}} = 0$, $\lambda_2 = 0$, $\lambda_3 = 0$ or $\lambda_4 = 0$, to see the influence of each loss on the performance of  DRR.

**Clarity**:
- The paper is overall well written and quite clear. However there is a problem with the labels of the axis and the legend of fig. 6.
- In Figure 3, you show examples of adversarial example reconstructed that's quite interesting. I'm also very curious about what the reconstruction $x'_{i,j}$ looks like during training, and believe it would be interesting to also plot it, to show if indeed the network is able to disentangle the features.
- You mention L210-211 that the hyperparameters are tuned using a binary search, it's not clear to me what you mean by that ?

**Significance**:
-  The proposed approach clearly outperforms other similar approach, however the approach introduces many losses and new hyperparameters without a lot of justification on why such loss are important. I believe this could prevent people from using this new method out-of the box, some ablation study could really help the paper have more impact.

**Time Spent Reviewing:**

4

---

> ### Author Response · Authors · 2021-08-10
> **Response to reviewer xJBn**
>
> ## Q1. Confidence interval and error bar.
> ## Q2. Legend of Fig. 6, definition $\hat{x}_i$, binary search of hyper-parameters and hinge loss.
> ## Q3. The purpose of hinge loss and ablation studies of multi hyper-parameters and losses.
> ## Q4. Plot the reconstruction output while training to show whether features are disentangled successfully.
>
> For Q1: we first note that we report our results according to FPR (x-axis) and TPR (y-axis), and the resultant ROC curve. This is a compact way for evaluating the binary classifier (adversarial detector in our case): larger area of the ROC, better classifier. In contrast, a confidence interval is an estimate, given an associated confidence level, calculated from the observed data (and the error bar is a bar chart of confidence intervals/deviations). The python package roc-utils provides a direct way to transform ROC curve to a set of different confidence intervals (associated with confidence levels). Due to page constraints, we will consider adding confidence intervals and error bars to the supplementary file for better visualization.
>
> For Q2: we note that Fig. 6 is correct as replied to the first reviewer's comments, it reports that the PGD attacker will become worse even the DRR defense mechanism is known to the attacker. And $\hat{x}_i$ is just a noisy version of $x_i$, as defined in the line above Eq. (1). For binary search, it means recursively searching for a hyper-parameter by dividing its associated range into two equal halves, the range that contains better results is used for the next iteration. We remark this is a commonly used strategy in adversarial detection [11, 23].
>
> For Q3: The hinge loss is to push the mimicked adversarial examples (counterexamples reconstructed from unpaired class and semantic features) from normal examples within a given range (controlled by the hyper-parameter $d$). For the comments on ablation studies, due to the page limit, we will add these studies to the supplementary file.
>
> For Q4: Similar to the reply to reviewer 1 Q1, we do not claim autoencoder can disentangle image features. Our disentangled features actually come from the parameter-fixed victim model's output label (class feature), and the encoder's output (semantic feature). The label and semantics of the same image are already disentangled representations.

---

### Official Review · Reviewer_8oh2 · 2021-07-20

**Rating:** 4
**Confidence:** 4

**Summary:**

This paper presents a method to detect adversarial examples by disentangling the representation. This is done by training autoencoders over both correct and incorrect class/semantic features. The authors claim that the proposed method outperformed other SOTA methods for adversarial detection and even works against adaptive adversarial attacks.


**Limitations And Societal Impact:**

The other question is related to the size of the dataset in experiment, we are using very small datasets i.e. MNIST and CIFAR 10 to train a GAN basically, and I am not sure whether the conclusion would generalize to large-scale datasets such as ImageNet which is 20x larger.

Last thing is about the adaptive attack which the authors added, why are the results on Cifar and MNIST different in terms of epsilon value? The description of Fig 6 is quite confusing, making me think PGD is a stronger attack than the adaptive attack, which shouldn’t be the case.


**Main Review:**

The core motivation of the paper is to distinguish robust features versus non-robust features using the DRR network. The assumption here is that the DRR could successfully disentangle the semantic features versus the class features. However, the establishment of such an assumption is still under question. There is no definition of the metric to measure disentanglment, we could show through statistical metrics such as Mutual information Gap (MIG), SAP score < Chen et al. 2019 Isolating Sources of Disentanglement in VAEs>.
- Especially in part 3 of the paper, there are many hand-wavy explanations, e.g. line 148. The idea of encoder/decoder structure is quoted vaguely, but it is used as a base logic. In a word, this cannot convince me that DRR is able to disentangle the features. In representation learning, disentanglement is still a very challenging problem as is pointed out by seminal work such as <Locatello et al. 2018> whereas there is no evidence showing that the proposed method could generate any type of compact representation.  (In a compact representation, a given factor is associated with only one or a few code dimensions. <Ridgeway et al. 2018 Learning Deep Disentangled Embeddings
With the F-Statistic Loss> From the resulting images in the experiment section and in the appendix, I just could not tell whether the semantic features are different from the class features.

**Time Spent Reviewing:**

4

---

> ### Author Response · Authors · 2021-08-10
> **Response to reviewer 8oh2**
>
> ## Q1. About the assumption of disentanglement with autoencoder.
> ## Q2. Applying to large-scale dataset.
> ## Q3. The parameter epsilon in adaptive attack and problem in Fig. 6.
>
> For Q1: we fully agree with the reviewer that disentanglement is a very challenging problem and there is no definite answer to this problem. However, our assumption is not "DRR could disentangle the semantic features versus class features", but DRR can make good use of class features and semantic features for adversarial detection, and class and semantic features are already disentangled in nature. The class features come from the outputs of the classification victim, which can correctly classify normal examples and wrongly classify adversarial examples with large probability. The semantic features come from the encoded representation from an autoencoder. For example, an adversarial 'dog' image (classified as 'cat' by the victim) will definitely have a class feature of a 'cat' and a semantic feature of a 'dog'. DRR just uses paired class/semantic features to mimic normal examples and unpaired class/semantic features to mimic adversarial examples under the self-supervised framework, but not to train a model to generate good disentangled features as commented. It is worth mentioning that in DRR, the class features come from the parameter-fixed victim, which cannot be trained at all.
>
> For Q2: regarding the scalability on a larger dataset, we have obtained the results over CIFAR100. We will add the result to the supplementary file, and leave the task of extending to more neural models and bigger datasets to the future work.
>
> For Q3: the accuracy changes with epsilon value is because epsilon controls the fidelity of the adversarial examples. The large epsilon is, the worse perceptual quality the adversarial examples have. Different datasets normally can tolerate different values of epsilon (e.g., CIFAR10 vs MNIST). Our setting is indeed the most commonly used values in the field of adversarial attack and defense [5, 12, 17].
> In Fig. 6, PGD appears to be stronger because there is no defense applied (i.e., non-adaptive attack). Adaptive PGD is weaker because it attacks the defended victim model (with DRR), and the non-adaptive PGD serves as the baseline to show DRR reduces the efficiency of PGD attack.

---

> > ### Comment · Reviewer_8oh2 · 2021-08-22
> > **Thank you for your replies!**
> >
> > Thank you for answering the questions!
> > I get what you are saying about disentanglement now. I think it would be better to rewrite this section about disentanglement and to limit the scope as we discussed. However, this kinda of ties back to my comment about scalability. What if the dataset is not about "cat" and "dog" which each has its own distinctive feature, but about some similar types of animals like "black bears" or "grizzly bears"? Base on my understanding of the paper, such datasets would adversely affect the performance of the proposed method.
> > All things considered, if the authors would provide additional results on larger-scale dataset, I would consider raising my score from 4 to 5.
> > Thanks!

---

> > > ### Author Response · Authors · 2021-08-26
> > > **Further response to reviewer 8oh2**
> > >
> > > Thanks for your valuable feedback, we will rewrite the section about disentanglement according to your suggestion in the final version. For the other insightful suggestion, we further evaluate the performance of DRR for detecting adversarial examples among classes/examples that share very similar semantic features. Specifically, we perform the experiment on dataset 'Stanford Dogs': http://vision.stanford.edu/aditya86/ImageNetDogs/, as a proof-of-concept study.
> > >
> > > This dataset is a subset of ImageNet and it has 120 dog breeds (i.e., 120 classes), each class contains about 150 images, and their semantic features are very similar from our view. For example: Affenpinscher (http://vision.stanford.edu/aditya86/ImageNetDogs/n02110627.html) v.s. Australian terrier (http://vision.stanford.edu/aditya86/ImageNetDogs/n02096294.html). Due to time constraints, we randomly choose 10 classes from this dataset. For each class, 100 images are then augmented by flipping, rotation, zooming and resizing to (64, 64, 3). As a result, we have 800 training images for each class, and ~50 test images for each class.
> > >
> > > The settings for adversarial attack and DRR defense are the same as those used for CIFAR-10 in the main manuscript, i.e., the victim model adopts the architecture of VGG16 and is trained by Adam (learning rate is 0.001). Under this setting, the AUC of ROC of DRR for detecting adversarial examples produced by PGD and FGSM (on the training examples) are as follows:
> > >
> > >
> > > **PGD**
> > >
> > > |$L_2 \ \epsilon=1$|$L_2 \ \epsilon=1.5$|$L_{inf} \ \epsilon=0.02$|$L_{inf} \ \epsilon=0.03$|
> > > |:-----:|:-----:|:-----:|:-----:|
> > > |0.8723|0.9032|0.9048|0.9131|
> > >
> > > **FGSM**
> > >
> > > |$L_2 \ \epsilon=1$|$L_2 \ \epsilon=1.5$|$L_{inf} \ \epsilon=0.02$|$L_{inf} \ \epsilon=0.03$|
> > > |:-----:|:-----:|:-----:|:-----:|
> > > |0.8747|0.8879|0.9073|0.9227|
> > >
> > > From the results, we can see that the performance of DRR in 'Stanford Dogs' is inferior to  CIFAR-10 in the main manuscript (similar to that of the Fashion dataset), but it is clear that DRR is still capable of detecting adversarial examples. And it is worth mentioning this performance drop is not because of the fact that all dog examples share very similar semantic features, but more due to the fact that VGG16 overfits this really small amount of data (800 for each class). Indeed, the victim model now has low test accuracy. To summarize, DRR can detect adversarial examples with similar semantic features. The complete results will be added to supplementary files.

---

### Decision · Program_Chairs · 2021-09-27

**Decision:**

Reject

**Comment:**

The manuscript introduces a defense against adversarial examples that relies on disentangling representations (here, disentanglement is not meant in the more common sense of feature disentanglement). On the methodology front, a number of concerns were raised about the ability of the approach to scale to larger datasets (where different classes are more likely to be difficult to disentangle). I also note that all experiments are performed on vision datasets. The current evaluation also fails to provide insights into the different hyperparameters introduced. An ablation study was suggested by the authors but not done in the current manuscript. Finally, the adaptive attack was not tailored to the proposed approach but rather simply evaluates an end-to-end attack. This is a good start but the authors did not provide a convincing argument for why it is sufficient in response to a comment raised in the reviews. In particular, given that the approach would involve a threshold should it be deployed in practice, I encourage the authors to provide evidence that a threshold offering sufficient accuracy on normal inputs while detecting adversarial inputs can be found - even in the presence of an adaptive adversary.